# CDC20-Mediated hnRNPU Ubiquitination Regulates Chromatin Condensation and Anti-Cancer Drug Response

**DOI:** 10.3390/cancers14153732

**Published:** 2022-07-31

**Authors:** Cindy Wavelet-Vermuse, Olena Odnokoz, Yifan Xue, Xinghua Lu, Massimo Cristofanilli, Yong Wan

**Affiliations:** 1Department of Pharmacology and Chemical Biology, Winship Cancer Institute, Emory University School of Medicine, Atlanta, GA 30322, USA; cindy.wavelet-vermuse@emory.edu (C.W.-V.); olena.odnokoz@emory.edu (O.O.); 2Department of Biomedical Informatics, University of Pittsburgh School of Medicine, Pittsburgh, PA 15206, USA; evasnow1992@hotmail.com (Y.X.); xinghua@pitt.edu (X.L.); 3Department of Medicine, Weill Cornell Medicine, New York, NY 10065, USA; mac9795@med.cornell.edu; 4Department of Hematology and Oncology, Winship Cancer Institute, Emory University School of Medicine, Atlanta, GA 30322, USA

**Keywords:** CDC20, hnRNPU, mitotic slippage, cohesin complex, chromatin condensation, drug resistance

## Abstract

**Simple Summary:**

CDC20 has been shown to function as an oncogene in multiple human malignancies and is recognized as a promising target for developing novel therapeutic strategies. Previous studies showed the role of CDC20 in cancer development and linked its function to drug resistance. Drug resistance can be acquired by multiple mechanisms, including chromatin organization and dynamic changes. It was previously reported that cancer cells with enlarged nuclei and diffuse chromatin showed higher resistance to anti-cancer therapy. We showed that CDC20 interacts with hnRNPU in the chromatin fraction and regulates its ubiquitination in breast cancer cells. CDC20-mediated hnRNPU ubiquitination modulates chromatin condensation by regulating the formation of the CTCF–cohesin complex. Dysregulation of the CDC20–hnRNPU axis results in diffuse chromatin and leads to drug resistance in breast cancer cells. Our findings shared new insights into the implication of the CDC20–hnRNPU axis in regulating chromatin organization and drug resistance, suggesting that the CDC20–hnRNPU axis could be a good target for cancer therapy.

**Abstract:**

Cell division cycle 20 (CDC20) functions as a critical cell cycle regulator. It plays an important role in cancer development and drug resistance. However, the molecular mechanisms by which CDC20 regulates cellular drug response remain poorly understood. Chromatin-associated CDC20 interactome in breast cancer cells was analyzed by using affinity purification coupled with mass spectrometry. hnRNPU as a CDC20 binding partner was validated by co-immunoprecipitation and immunostaining. The molecular domain, comprising amino acid residues 461–653, on hnRNPU required for its interaction with CDC20 was identified by mapping of interactions. Co-immunoprecipitation showed that CDC20-mediated hnRNPU ubiquitination promotes its interaction with the CTCF and cohesin complex. The effects of CDC20–hnRNPU on nuclear size and chromatin condensation were investigated by analyzing DAPI and H2B-mCherry staining, respectively. The role of CDC20–hnRNPU in tumor progression and drug resistance was examined by CCK-8 cell survival and clonogenic assays. Our study indicates that CDC20-mediated ubiquitination of hnRNPU modulates chromatin condensation by regulating the interaction between hnRNPU and the CTCF–cohesin complex. Dysregulation of the CDC20–hnRNPU axis contributes to tumor progression and drug resistance.

## 1. Introduction

CDC20 is a crucial regulator of anaphase-promoting complex/cyclosome (APC/C), which regulates the initiation of anaphase and the exit from mitosis through time-dependent degradation of securin and cyclin B [1,2]. Beyond its mitotic function, the role of CDC20-APC/C has been documented to regulate a series of cellular processes, including apoptosis, brain development, cancer stem cell proliferation, and genome stability [3,4,5,6]. Dysfunction of CDC20 leads to aneuploidy that contributes to various human diseases or predisposes normal cells to become cancerous [7,8]. Recent Cancer Genome Atlas (TCGA) and human specimen-orientated pathological cohort studies have demonstrated a strong correlation between the uncontrolled accumulation of CDC20 with poor prognosis of cancers [9,10,11]. We and others have observed that elevated CDC20 expression was often detected in tumor tissues, including breast, colorectal, lung, bladder, cervical, liver, ovarian, and prostate cancers. Results from pathological analysis further suggested the overexpression of CDC20 and securin as the hallmark of triple-negative breast cancer (TNBC) with a short survival period for breast cancer patients [10]. In addition, recent therapeutic studies have pointed out the critical role of CDC20 in drug resistance although the underlying molecular mechanisms remain largely unknown. Thus, uncovering the mystery by which aberrant accumulation of CDC20 drives drug resistance could lead to a new therapeutic strategy for cancer control.

Antimitotic agents such as microtubule inhibitors (paclitaxel, docetaxel) are widely used in cancer therapy. These agents impose a prolonged mitotic arrest in cancer cells that rely on sustained activation of the SAC (spindle-assembly checkpoint) and may lead to subsequent apoptosis [12]. Continual exposure of cancer cells to mitotic drugs leads to the development of cellular drug resistance, but the mechanisms remain unclear. It was thought that mitotic slippage might serve as one of the mechanisms for drug resistance, where cancer cells fail to be arrested in response to the mitotic blockade resulting in the escape from mitosis to the next interphase [13,14,15,16]. Despite conditions that activate the SAC, CDC20-APC/C is activated upon mitotic slippage of cancer cells to G1 phase cells [17]. The possible correlation between the elevated CDC20 and mitotic slippage was observed in different types of cancer cells [18,19]. Moreover, depletion of CDC20 counteracts malignancy progression and drug resistance [20,21,22]. Pharmacological inhibition of CDC20-APC/C by either CDC20 inhibitor Apcin or CDC20 PROTAC CP5V blocks mitotic progression and induces tumor cell death [23,24]. On the other hand, recent high-resolution microscopy studies have revealed the correlation between cellular drug resistance with abnormal chromatin dynamics. Increased chromatin packing density fluctuations could facilitate tumorigenesis and allow cancer cells to develop chemotherapeutic resistance [25]. However, whether CDC20 plays a role in regulating chromatin behavior and whether its deregulation is correlated with drug resistance remains poorly understood.

hnRNPU (heterogeneous nuclear ribonucleoprotein U) is a DNA- and RNA-binding protein involved in several cellular processes such as telomere-length regulation [26], transcription [27], mRNA alternative splicing and stability [28], Xist-mediated transcriptional silencing [29], and mitotic cell progression [30]. Structurally, hnRNPU comprises a low complexity RNA-binding RGG repeat and an ATP-binding AAA+ domain, known to facilitate the assembly and operation of diverse proteins and nucleoprotein machines [31]. More recently, hnRNPU has been reported to act as a major factor in maintaining 3D chromatin architecture and regulating the formation of chromatin loops [32,33]. The strengths of chromatin loops are reduced upon hnRNPU depletion [32]. hnRNPU is involved in regulating the cohesin complex via its association with CTCF and RAD21 [32]. In addition, hnRNPU oligomers could interact with chromatin-associated RNAs to regulate interphase chromatin structures in a transcription-dependent manner [33]. Malfunction of hnRNPU and misregulated chromatin loop formation are linked to carcinogenesis. hnRNPU interacts with CTCF resulting in transcriptional alteration of genes associated with tumor progression [34]. Song et al. showed that binding of long noncoding RNA *HNF4A-AS1* to the RGG domain of hnRNPU facilitates hnRNPU–CTCF interaction leading to CTCF transactivation and subsequent transcriptional alteration of *HNF4A* and other cancer-associated genes in neuroblastoma (NB) cells [34]. Moreover, hnRNPU promotes breast cancer cell proliferation, migration, and invasion [35]. Therefore, investigation of hnRNPU regulation in breast cancer circumstances and the development of related targeting strategies could lead to a new anticancer therapy.

In the current study, we report the previously unknown role of CDC20 in regulating chromatin condensation. Using protein complex purification coupled with mass spectrometry analysis, we have identified hnRNPU as a chromatin-associated binding partner for CDC20. CDC20-mediated hnRNPU ubiquitination facilitates the formation of the CTCF–cohesin complex, suggesting that CDC20–hnRNPU modulates chromatin loop formation and transcription. Uncontrolled elevation of CDC20 impedes the homeostasis of chromatin dynamics resulting in chromatin diffusion via the CDC20–hnRNPU axis that could promote cellular drug resistance. Thus, the blockade of CDC20 chromatin-associated functions could be a new targeting strategy for drug resensitization.

## 2. Materials and Methods

### 2.1. Cell Lines and Culture Condition

MCF-10A, DCIS, MDA-MB-231, MDA-MB-468, HCC1937, SK-BR-3, AU-565, HCC1954, BT474, MCF7, T47D, and 293T cells were obtained from the American Type Culture Collection (ATCC). All cell lines were cultured at 37 °C under 5% CO_2_ in the appropriate medium containing 10% fetal bovine serum (FBS), 100 U mL^−1^ penicillin sodium, and 100 μg mL^−1^ streptomycin sulfate, according to the provider’s recommendations.

### 2.2. Western Blot

Proteins were extracted from cells in RIPA lysis buffer with protease inhibitors (S8820, Sigma, Saint-Louis, MO, USA). Antibodies against CDC20 (1:1000; sc-13162, Santa Cruz, Dallas, TX, USA), hnRNP-U (1:1000; sc-32315), pH3 (1:1000; 06-570; EMD Millipore, Burlington, MA, USA), MPM-2 (1:1000; 05-368; EMD Millipore), cyclin B1 (1:1000; sc-245), cyclin A (1:1000; sc-271682), cyclin E (1:1000; sc-247), PARP (1:1000; 9532; CST), GAPDH (1:1000; NB300-221, Novus Biologicals, Centennial, CO, USA), HSP90 (1:1000; 4877, CST, Beverly, MA, USA), H3 (1:1000; 4499; CST), Myc (1:1000; sc-40), Ubiquitin (1:1000; 43124, CST), CTCF (1:1000; 3418; CST), RAD21 (1:1000; ab992; Abcam, Boston, MA, USA), V5 (1:2000; 80076; CST), Flag (1:1000; 14793; CST), and HA (1:1000; 3724; CST) were used in this study. Antibody against actin (1:5000; A5441, Sigma) was used as a loading control.

### 2.3. Bioinformatics Analysis

The bioinformatic portal UALCAN (http://ualcan.path.uab.edu, accessed on 25 January 2022) was used to analyze breast cancer transcriptome and proteome data. The resource uses The Cancer Gene Atlas (TCGA) and the Clinical Proteomic Tumor Analysis Consortium (CPTAC) databases [36]. We analyzed the expression of CDC20, genome stability-, chromosome segregation-, mitotic exit-, and Cdc20-APC/C pathway-related genes in several breast cancer subtypes and built heat maps. CDC20 and hnRNPU mRNA and protein expression were also analyzed among several types of breast cancer using UALCAN. A *p*-value of less than 0.05 (*p* < 0.05) was considered statistically significant.

### 2.4. Plasmid Constructions and Mutagenesis

HA-CDC20 (Plasmid #11594) and pcDNA3.1-hnRNPU-V5 (Plasmid #35974) plasmids were ordered from Addgene. All short hairpin RNA (shRNA) targeting CDC20 or hnRNPU were purchased from Sigma Aldrich and the negative control vector pLKO.1 puro was purchased from Addgene (#8453).

pHAGE-CDC20 and plenti6/V5-hnRNPU plasmids were generated by PCR amplification followed by subcloning into pENTR-D-TOPO (Invitrogen, Waltham, MA, USA) to obtain the entry clone. LR recombination was performed to transfer the CDC20 or hnRNP-U gene from the entry construct into a Gateway destination vector pHAGE or pLenti6/V5-DEST, respectively.

hnRNP-U fragments were amplified by PCR and subcloned into the pKMyc vector. CDC20 full length was subcloned into the pIRES vector with FLAG/HA tag. All plasmids were confirmed by sequencing.

hnRNPU 4KR mutations were generated using the QuickChange XL Site-Directed Mutagenesis kit (catalog No. 200517, Agilent, Santa Clara, CA, USA). Mutations were confirmed by sequencing.

### 2.5. Lentiviral Particles Production and Infection

Lentiviral particles were produced by transfecting the expression plasmid (shRNA, pHAGE-CDC20, pLenti6/V5-hnRNPU, and pLenti6-H2B-mCherry) together with pRRE, pRSV-Rev, and pVSVg packaging plasmids into 293T cells using polyethylenimine (PEI MAX, 24765, Polysciences Inc., Warrington, PA, USA). Viruses were harvested 24 and 48 h post-transfection and filtered through 0.45 µm filters. MDA-MB-231 or MDA-MB-468 cells were transduced with lentiviruses in the presence of polybrene (H9286; Sigma). After infection, the stable cell lines were established by selection with 1 µg/mL of puromycin or 10 µg/mL of blasticidin.

### 2.6. Survival Assay

Cells were seeded in a 96-well plate at a density of 5000 cells per well and pre-incubated at 37 °C for 24 h. Cells were treated with different concentrations of docetaxel for 48, 72, or 96 h. The medium was replaced with 100 µL of DMEM added by 10% of Cell Counting Kit-8 (CCK-8) solution (Dojindo, Rockville, MD, USA) and incubated for 1 h at 37 °C. The absorbance was measured at 450 nm using a microplate reader.

### 2.7. Microscopy, Immunofluorescence, and Images Analysis

Cells seeded on coverslips were fixed with 4% paraformaldehyde in PBS for 15 min. Cells were permeabilized by incubating with 0.1% Triton X-100 in PBS for 15 min on ice. After blocking for 1 h, the appropriate antibodies were added and incubated overnight at 4 °C. The cells were washed with PBS and incubated with Alexa 488 goat anti-mouse IgG (A11029, Invitrogen) and/or Alexa 594 goat anti-rabbit IgG (A32740, Invitrogen) diluted in blocking buffer for 1 h. After washes, the cells were incubated with DAPI (Catalog No. 62248, Thermo Scientific, Waltham, MA, USA), and the samples were mounted by using a mounting medium (H-1700, Vectashield, Vector Laboratories, Inc., Newark, CA, USA). The images were taken under the Leica SP8 confocal microscope with the LASX software (Deerfield, IL, USA).

The areas of DAPI-stained nuclei were analyzed by using FIJI software (Madison, MA, USA).

The percentage of condensed chromatin in cells transduced with pLenti6-H2B-mCherry was evaluated with FIJI by measuring the area of the nuclei, setting up a threshold, and measuring the area corresponding to the condensed chromatin.

### 2.8. Flow Cytometry

After pretreatment with 100 nM docetaxel (catalog No. T1034, TargetMol, Wellesley Hills, MA, USA) for 20 h, MDA-MB-468 cells were treated with 50 nM of either hesperadin (Catalog No. S1529, Selleckchem, Houston, TA, USA) or DMSO for different times in the continual presence of 100 nM docetaxel. Single-cell suspensions fixed with ice-cold ethanol were stained with propidium iodide staining solution and analyzed by flow cytometry.

### 2.9. Crosslinking and Cell Fractionation

Before fractionation, cells were crosslinked with dithiobis (succinimidyl propionate) (DSP, catalog No. 22585, Thermo Fisher, Waltham, MA, USA) or disuccinimidyl suberate (DSS, catalog No. 21655, Thermo Fisher). Cell fractionation was performed as described [37,38]. Briefly, cells were lysed in Buffer A (10 mM HEPES-KOH, pH 7.9, 10 mM KCl, 1.5 mM MgCl_2_, 340 mM sucrose, 10% glycerol, Protease inhibitors, 10 mM sodium butyrate, 0.1% Triton X-100). The nuclei were pelleted, resuspended in Buffer A containing 300 mM of NaCl, and centrifuged. The chromatin pellet was resuspended in Buffer A and digested with micrococcal nuclease (New England Biolabs, Ipswich, MA, USA, M0247S) for 15 min at 37 °C. Debris was pelleted by centrifugation at 20,000× *g* for 20 min.

### 2.10. Purification and Mass Spectrometry

MDA-MB-468 cells stably expressing Flag/HA-CDC20 were fractionated to obtain the chromatin fraction. CDC20-interacting proteins were purified by immunoprecipitation with anti-FLAG M2 beads (Catalog No. A2220, Sigma-Aldrich, Saint-Louis, MO, USA). Beads were washed four times with TBS buffer and the complexes were eluted with 3X FLAG peptide (Catalog No. F4799, Sigma-Aldrich) in TBS buffer. The elution was then separated on SDS-PAGE followed by silver staining (catalog No. 24612, Pierce, Waltham, MA, USA) or run into the stacking gel and cut out for mass spectrometry analysis.

### 2.11. Immunoprecipitation

Cells were lysed in ice-cold RIPA buffer supplemented with protease inhibitors at 4 °C. The cell lysates were centrifuged at 12,000× *g* for 20 min. The protein concentrations were determined using the BCA Protein Assay kit (catalog No. 23225, Pierce). The cell lysates were incubated with the appropriate antibodies overnight at 4 °C on a rotator. A/G-agarose beads (sc-2003, Santa Cruz) were added to the reaction and incubated for 4 h at 4 °C on a rotator. After four washes with RIPA buffer, the beads were boiled in 1× electrophoresis sample buffer for 5 min.

### 2.12. Clonogenic Assay

Cells were seeded in 6-well plates and preincubated at 37 °C for 24 h. Cells were treated with different concentrations of docetaxel for 24 h. The medium was replaced with fresh medium. Cells were cultured for additional 16 days in a drug-free medium. Colonies were fixed and stained with methanol/crystal violet solution.

### 2.13. Statistical Analysis

Results were reported as mean ± SD (standard deviation). Statistical analysis was performed using Microsoft Excel. Statistical significance between two groups was calculated by using a Student’s *t*-test. A *p*-value of < 0.05 was considered significant.

## 3. Results

### 3.1. Elevated CDC20 Correlates with Poor Breast Cancer Prognosis and Drug Resistance

To examine the possible clinical relevance of CDC20 in breast cancer development and drug resistance, we measured the protein expression levels of CDC20 in the normal breast epithelial (MCF-10A), the ductal carcinoma in situ (DCIS), and different breast cancer cell lines, including TNBC, HER2+, and Luminal subtypes (Figure 1a). While CDC20 expression is relatively low or moderate in normal breast epithelial cells, DCIS, HER2+, and Luminal cells, a significant accumulation of CDC20 is observed in TNBC cells, particularly in MDA-MB-231 and MDA-MB-468. Moreover, we compared the expression profiles of CDC20 with a series of genome stability-, chromosome segregation-, mitotic exit-, and CDC20-APC/C pathway-related genes between different breast cancer subtypes and normal tissues (Figure 1b and Appendix A). The expression levels of CDC20 in breast cancer tissues are significantly higher than in normal tissues. Furthermore, higher expression levels of CDC20 were observed in TNBC compared with other breast cancer subtypes (Figure 1b and Appendix A). The protein expression levels of CDC20 in TNBC are also significantly higher than in Luminal and HER2+ subtypes of breast cancer (Appendix A). The high mRNA expression of CDC20 was associated with a poor overall survival time in breast cancer patients (Figure 1c). To study whether CDC20 accumulation is involved in the mechanism of drug resistance developed by breast cancer cells, we analyzed CDC20 knockdown or overexpression in two TNBC cell lines, MDA-MB-231 and MDA-MB-468 (Figure 1d,e). Depletion of CDC20 in TNBC cells increased response to docetaxel treatment (Figure 1f). In contrast, overexpression of CDC20 decreased the response to docetaxel treatment (Figure 1f). These results demonstrate a strong connection between aberrant elevation of CDC20 expression, poor breast cancer prognosis, and drug resistance. These results further suggest that targeting CDC20 could be a good strategy for antimitotic therapy.

### 3.2. Elevated CDC20 Is Correlated with Mitotic Slippage, a Possible Reason Causing an Increase in Nuclear Size and Chromatin Decondensation in Cancer Cells

As shown previously, breast cancer cells treated with docetaxel are arrested in mitosis [39]. The prolonged mitosis arrest can lead to cell death. However, some cells can slip out of mitosis and re-enter the G1 phase without cytokinesis by a mechanism called mitotic slippage [15]. This phenomenon is mainly observed in TNBC cells. Given the critical role of CDC20 in governing mitotic progression, we asked whether elevated CDC20 is involved in the mechanism of mitotic slippage in TNBC cells. The immunoblot results demonstrated that MDA-MB-468 cells overexpressing CDC20 exit mitosis earlier in response to docetaxel treatment, as shown by pH3 and MPM-2 mitotic markers and by the degradation of Cyclin B1 (Figure 2a top panel). So, CDC20 facilitates the exit of mitosis induced by docetaxel. However, we did not observe significant differences in the pH3, MPM-2, and Cyclin B1 levels between control cells and cells expressing CDC20 knockdown (Figure 2a top panel). This can be explained by the fact that docetaxel induces mitotic arrest by activating of spindle assembly checkpoint (SAC) which incorporates anaphase promoting complex (APC/C) coactivator CDC20 into the mitotic checkpoint complex (MCC) and prevents the activation of the APC/C. Likewise, the deletion of CDC20 inhibits the activation of APC/C. Since docetaxel treatment and CDC20 knockdown affect the same APC/C-mediated mitotic pathways, we did not observe significant differences in the levels of mitotic markers, such as pH3, MPM-2, and Cyclin B1. It was previously reported that mitotic arrest induced by docetaxel and other taxanes requires the activity of Aurora B kinase to maintain the active SAC. Aurora B kinase inhibitors disrupt SAC and facilitate mitotic slippage promoting resistance of taxanes. To induce mitotic slippage, we treated cells with docetaxel, followed by the treatment of hesperadin, an Aurora B kinase inhibitor.

Flow cytometry showed that docetaxel treatment arrests the cells in the G2/M phase (Figure 2b, Appendix A). The addition of hesperadin allows the cells to exit mitosis and re-enter the G1 phase with 4N DNA (Figure 2a, Appendix A). The addition of hesperadin enhances the mitotic slippage phenomenon in CDC20 overexpressing cells (Figure 2a bottom panel). We were wondering if the cells exit mitosis and die or if they enter a new cell cycle with 4N DNA. To answer this question, we measured the levels of full- and cleaved-PARP protein. The results indicate that, even if some portion of cells undergo apoptosis, as illustrated by cleaved-PARP protein, the overexpression of CDC20 allowed the cells to exit mitosis and proliferate, as shown by the levels of full-PARP protein (Figure 2a bottom panel). In contrast, the knockdown of CDC20 induces apoptotic cell death as shown by the cleaved-PARP levels (Figure 2a bottom panel). Using CCK8 cell survival assay, we confirmed that control or overexpressing CDC20 TNBC cells slip out mitosis and continue to proliferate. However, TNBC cells expressing CDC20 knockdown undergo apoptosis (Appendix A). These data suggest that elevated CDC20 allows cells to exit mitosis without proper chromosome segregation.

Cells that slip out of mitosis present an enlarged nucleus [15]. Moreover, drug resistance has been linked with an abnormal increase in the nucleus size and changes in chromatin behavior [40]. However, the mechanisms by which diffuse chromatin induces drug resistance are still unclear. Thus, we decided to evaluate whether CDC20 is involved in diffusion/decondensation of the chromatin and if CDC20 expression levels are associated with changes in the nucleus size of TNBC cells. DAPI staining analysis showed that the size of the nuclei significantly decreased (from 130.1 µm to 91.4 µm) after the CDC20 knockdown (Figure 2c and Appendix A). In contrast, the nuclei size significantly increased (from 130.1 µm to 154.5 µm) after CDC20 overexpression (Figure 2c). Moreover, we measured the percentage of condensed chromatin in MDA-MB-468 cells with CDC20 knockdown or overexpression transduced with H2B-mCherry. Indeed, the TNBC cells with CDC20 knockdown had more condensed chromatin (Figure 2d and Appendix A).

These results show the role of CDC20 in mitotic slippage, an increase in nuclear size, and chromatin condensation. 

### 3.3. Identification of Nuclear Matrix Protein hnRNPU as a Target for CDC20 during Chromatin Dynamics

To understand the role of CDC20 in chromatin behavior, we measured the expression levels of chromatin-associated CDC20 through the cell cycle. For this purpose, we synchronized the cell cycle of TNBC cells by double thymidine treatment and fractionated them into the cytosol, nuclear, and chromatin fractions (Figure 3a). The efficiency of cell cycle synchronization was determined by Western blot using antibodies against cyclins A, B, and E, and a mitotic marker pH3 (Figure 3b). To our surprise, we observed for the first time that high CDC20 levels were detected in the chromatin fraction at 6 and 12 h after release from thymidine, corresponding to S and M phases, respectively (Figure 3c). These results suggest a novel role for CDC20 in association with chromatin regulation.

To understand how CDC20 is involved in changes in chromatin behavior, we purified chromatin-associated Flag/HA-CDC20 complexes from cells synchronized in the S phase and analyzed CDC20 interactome by mass spectrometry [41]. Mass spectrometry interactome analysis revealed proteins that play an important role in the control of cell cycle (CDK1), histone proteins (H2A, H4, H2B, and H3), as well as the heterogeneous nuclear ribonucleoprotein U (hnRNPU) (Figure 3d,e). As previously described, hnRNPU plays a critical role in regulating chromatin dynamics through functional interaction with cohesion complex and chromatin loop formation [32,33]. Dysregulation of hnRNPU results in carcinogenesis and drug resistance [34,35,42]. Co-immunoprecipitation assays performed in MDA-MB-468 cells confirmed the interaction between endogenous CDC20 and hnRNPU (Figure 3f). Furthermore, immunostaining followed by confocal microscopy revealed the colocalization of CDC20 and hnRNPU in the nucleus (Figure 3g). These results showed that CDC20 is associated with chromatin and suggested that CDC20 may play an important in regulating chromatin unpacking through interaction with hnRNPU.

### 3.4. High Expression of hnRNPU in Breast Cancer and Mapping of Molecular Interaction between CDC20 and hnRNPU

The impact of hnRNPU in carcinogenesis has been documented previously in different types of tumors [34,42]. To determine the role of hnRNPU in breast cancers, we measured the expression of hnRNPU in normal breast epithelial, DCIS, and different breast cancer cell lines, including TNBC, HER2+, and Luminal subtypes. As shown in Figure 4a, the expression pattern for hnRNPU seems to be the same as for CDC20. Additionally, a strong correlation between CDC20 and hnRNPU expression in breast cancer patients was shown by using correlationAnalyzerR (Figure 4b) [43]. Moreover, Figure 4c,d indicate that the expression levels of hnRNPU in human breast cancer tissues are significantly higher than in normal tissues. The high mRNA expression of hnRNPU was associated with a poor overall survival time in breast cancer patients (Figure 4e).

To further identify which region of hnRNPU is essential for its interaction with CDC20, we constructed a series of Flag/HA-tagged CDC20 and Myc-tagged hnRNPU truncation mutants and performed co-IP experiments (Figure 4f,g). Co-IP analysis revealed that the region comprising amino acid residues 461–653 and corresponding to the AAA+ domain in hnRNPU is required for its interaction with CDC20 (Figure 4h). These results suggest that CDC20 binds the AAA+ domain of hnRNPU, and this interaction may be responsible for regulating chromatin behavior and breast cancer progression.

### 3.5. CDC20 Regulates hnRNPU Ubiquitination and Its Interaction with CTCF

To understand the role of CDC20–hnRNPU interaction in the chromatin fraction in TNBC cells, we first analyzed whether CDC20 regulates the stability of hnRNPU protein. We treated the TNBC cells with cycloheximide, a protein synthesis inhibitor. Surprisingly, either CDC20 knockdown or CDC20 overexpression does not affect the protein stability of hnRNPU (Appendix A). Additionally, the treatment of TNBC with MG132 (a proteasomal inhibitor) or chloroquine (an autophagy inhibitor) did not significantly affect the hnRNPU expression levels (Appendix A). Next, we tested whether CDC20 regulates ubiquitination of hnRNPU in TNBC cells using pull-down experiments. As shown in Figure 5a,b, the knockdown of CDC20 prevents the ubiquitination of hnRNPU. In contrast, elevated CDC20 expression led to increased ubiquitination of hnRNPU (Figure 5a,b).

It was previously reported that hnRNPU interacts with CTCF and RAD21 [32], two proteins involved in the formation of chromatin loops. Thus, we decided to investigate if CDC20 is required for the interaction between hnRNPU, CTCF, and RAD21. We pulled down CTCF from CDC20 knockdown or overexpressing TNBC cells. The results showed that CDC20 knockdown prevents the interaction between CTCF and hnRNPU and between CTCF and RAD21 (Figure 5c). In contrast, CDC20 overexpression increased the interaction between CTCF and hnRNPU and between CTCF and RAD21 (Figure 5c). The results suggest that CDC20 may play a regulatory role in 3D chromatin organization through regulation of the interaction between hnRNPU, CTCF, and RAD21 proteins. To study whether CDC20-mediated ubiquitination of hnRNPU is required for its interaction with CTCF and RAD21, we predicted the ubiquitin sites on hnRNPU by using the PhosphoSitePlus web resource. Next, we mutated the four predicted ubiquitin sites at lysine (K) residues K352, K464, K524, and K565 of hnRNPU. Lysine to arginine mutation of the four residues (4KR: K352R, K464R, K524R, and K565R) significantly decreases the ubiquitination of hnRNPU (Figure 5d). We showed that the mutation of the four ubiquitin sites reduces the interaction between CDC20 and hnRNPU (Figure 5e) and between CTCF and hnRNPU (Figure 5f). These results suggest that hnRNPU interaction with CDC20 and its ubiquitination are required for hnRNPU–CTCF-RAD21 complex formation and regulation of chromatin loop formation.

### 3.6. Inhibition of CDC20-Mediated hnRNPU Ubiquitination Results in the Decreased Nuclear Area, Enhanced Chromatin Condensation and Cellular Sensitization to Docetaxel Treatment

To examine the physiological relevance of CDC20-mediated ubiquitination of hnRNPU, we measured a series of physiological alterations and hnRNPU ubiquitination. First, we analyzed the nucleus area and chromatin condensation of MDA-MB-468 cells expressing hnRNPU knockdown (Appendix A) or overexpressing hnRNPU (Appendix A). DAPI staining analysis showed that the area of the nuclei significantly decreased (from 114.2 µm^2^ to 103.2 µm^2^, *p* < 0.05) after hnRNPU knockdown (Figure 6a and Appendix A). In contrast, the nucleus area significantly increased (from 114.2 µm^2^ to 129.6 µm^2^, *p* < 0.01) after hnRNPU overexpression (Figure 6a). Moreover, analysis of H2B-mCherry fluorescence showed that TNBC cells with hnRNPU knockdown have more condensed chromatin while cells with hnRNPU overexpression have less condensed chromatin (Figure 6b). Next, we analyzed the effect of hnRNPU on the ability of TNBC cells to form colonies. Interestingly, TNBC cells with knockdown of hnRNPU were unable to form colonies (Figure 6c) and sensitized TNBC cells to docetaxel treatment (Figure 6d). In contrast, hnRNPU overexpression induced colony formation and higher resistance to docetaxel treatment (Figure 6d,e). Additionally, we mentioned that TNBC cells overexpressing hnRNPU grow faster than control cells. In contrast, TNBC cells expressing hnRNPU knockdown grow slower than control cells (Appendix A). To assess the physiological impact of disruption of hnRNPU ubiquitination, we expressed hnRNPU knockdown in MDA-MB-468 cells followed by the rescue of hnRNPU-WT or hnRNPU-4KR (four lysines residues on hnRNPU were mutated into arginine residues) (Figure 6e). The rescue of hnRNPU-WT led to a significant increase in the nucleus area (from 138.1 µm^2^ to 157.6 µm^2^, *p* < 0.05) (Figure 6f) and more diffuse chromatin (Figure 6g). The rescue of hnRNPU-WT in TNBC cells with hnRNPU knockdown significantly promoted the formation of colonies (Figure 6h) and increased the resistance of cells to docetaxel treatment (Figure 6d), while the rescue of hnRNPU-4KR did not show a significant effect on colony formation (Figure 6h) and cell survival under docetaxel treatment (Figure 6d) in hnRNPU knockdown cells. These results suggest that disruption of CDC20-mediated hnRNPU ubiquitination enhances the chromatin condensation and sensitizes the cells to docetaxel treatment. Finally, we measured the levels of several drug resistance-, tumorigenesis-, and mitotic-related proteins after rescuing hnRNPU-WT or hnRNPU-4KR (Figure 6i and Appendix A). We showed that the rescue of hnRNPU-WT increases the level of drug resistance- or tumorigenesis-related proteins, such as CDK7 [44,45] or FoxM1 [46] (Figure 6i). Thus, the modulation of the CDC20–hnRNPU axis could improve the chromatin condensation in TNBC, which in turn resensitizes cellular drug response.

## 4. Discussion

The role of CDC20 in regulating mitotic progression [1,2], brain development [47,48], apoptosis [5,49], and tumorigenesis [50,51,52] has been thoroughly studied. However, little is known about how CDC20 is involved in drug resistance. For the first time, we report that CDC20 physically interacts with chromatin and modulates chromatin condensation (Figure 7). Based on an unbiased proteomic analysis, we identified hnRNPU as an interacting partner for CDC20. Interestingly, we showed that CDC20-mediated hnRNPU ubiquitination governs drug resistance by regulating the formation of the CTCF–RAD21 complex (Figure 7). Disruption of CDC20-mediated hnRNPU ubiquitination reduces cell growth and sensitizes TNBC cells to antimitotic chemotherapeutic drugs. This study unveils a new view for CDC20 in regulating chromatin dynamics and uncovers the molecular mechanism by which CDC20 modulates drug resistance, which provides a novel strategy for developing anti-cancer therapy.

hnRNPU is involved in the decondensation of the chromatin during interphase [33]. Moreover, hnRNPU interacts with CTCF and/or RAD21, and it may affect a variety of their functions, including chromatin looping, 3D chromatin organization, and transcriptional regulation [32]. The cooperation between hnRNPU and CTCF in maintaining 3D chromatin organization was previously reported in mouse hepatocytes [32] and human NB cells [34]. Mechanistically, we found that in breast cancer CDC20-mediated hnRNPU ubiquitination regulates the formation of the CTCF–RAD21 complex, two proteins involved in the formation of chromatin loops [53,54]. A previous study has demonstrated that the knockdown of CTCF and RAD21 compact the chromatin. Compaction can be explained by the reduction in the chromatin loop number [55]. We showed that hnRNPU depletion and prevention of hnRNPU ubiquitination significantly decreased the size of the nuclei and resulted in more condensed chromatin. These results suggest that preventing the CDC20-mediated hnRNPU ubiquitination abolishes the formation of the CTCF–RAD21 complex and subsequently prevents the chromatin looping and decondensation. Furthermore, reduction in the chromatin loop number would allow the compaction of the chromatin. However, further studies are required to better understand how CDC20–hnRNPU impacts the chromatin organization. A recent study showed that hnRNPU mainly binds to active chromatin and the majority of hnRNPU peaks locate at promoters and overlap with CTCF and/or RAD21 [32]. Since ubiquitination is a PTM with several regulatory roles including proteasomal degradation and protein trafficking, we tested whether CDC20-mediated hnRNPU ubiquitination is the mechanism that targets hnRNPU for degradation. We did not see a significant difference in hnRNPU levels in breast cancer cells with CDC20 overexpression or knockdown. These data suggest that CDC20-mediated ubiquitination of hnRNPU does not target hnRNPU for protein degradation but may play an important regulatory role in the interaction between hnRNPU and chromatin and between hnRNPU and other proteins, such as CTCF or RAD21, to regulate chromatin loop formation and 3D chromatin organization. However, additional experiments are required to test this hypothesis. Chromatin immunoprecipitation-sequencing (ChIP-seq) analysis of hnRNPU, CTCF, and RAD21 binding in TNBC expressing CDC20 knockdown could elucidate if CDC20–hnRNPU promotes the recruitment of CTCF and RAD21 on chromatin. To investigate the impact of CDC20–hnRNPU on chromatin interaction (compartments, topologically associating domains, and chromatin loops), performing Hi-C assays would be required. Folding the chromosomal fiber in the interphase is an important element regulating gene expression [56,57,58]. In particular, physical contacts between promoters and enhancers are necessary for cell-type-specific transcription [59]. RNA-seq analysis would be useful to understand how the disruption of CDC20–hnRNPU alters gene expression.

CDC20 is well-characterized as an oncogene and was shown to be highly expressed in several human cancers [11]. High expression of CDC20 is a poor prognosis marker and is associated with cancer progression, metastasis, and resistance to therapy [52,60]. These results pinpointed the feature of CDC20 to be an ideal clinical target. Great efforts have been made to develop small-molecule inhibitors of CDC20 that block APC/C^Cdc20^ function in the past decade [24]. Randall King’s group has developed several CDC20 small molecule inhibitors and tested their role in tumor growth inhibition in various preclinical models [24,61,62]. Future clinical studies and chemical modifications will enhance the application of CDC20 inhibitors in anti-cancer treatment. More recently, using PROTAC technology, our lab developed a chimera compound called CP5V that inhibits mitotic progression [23]. CP5V specifically degrades CDC20 by bridging CDC20 to the VHL/VBC complex for ubiquitination-mediated degradation, suppressing breast tumor progression [23]. While the development of CDC20 inhibitors provides a new avenue for antimitotic therapy through blockade of the mitotic progression, the identification of new functions of CDC20 in regulating chromatin dynamics and drug resistance via the CDC20–hnRNPU axis offers an innovative perspective to target CDC20-mediated drug resistance.

hnRNPU and CTCF have been described to regulate target genes affecting cancer development. In NB cells, hnRNPU interacts with CTCF resulting in transcriptional alteration of genes associated with tumor progression [34]. Moreover, hnRNPU expression was associated with cisplatin sensitivity in bladder cancer [42]. Additionally, hnRNPU has been shown to promote breast cancer cell proliferation, migration, and invasion [35]. CTCF can also affect breast cancer development by regulating target genes [63]. We showed that the expression levels of hnRNPU in human breast cancer tissues are significantly high. The high protein expression of hnRNPU was associated with a poor overall survival time in breast cancer patients. Moreover, the knockdown of hnRNPU in TNBC cells prevents cell growth and sensitizes the cells to docetaxel treatment. These results suggest that hnRNPU may act as an oncogene. Further determination of the physiological impact of hnRNPU in breast carcinogenesis and anti-cancer drug resistance using preclinical breast cancer models will confirm our documented oncogenic effect for hnRNPU in breast cancer.

## 5. Conclusions

In conclusion, we demonstrated that CDC20-mediated hnRNPU ubiquitination modulates chromatin condensation by regulating the interaction between hnRNPU and the CTCF–cohesin complex. Dysregulation of the CDC20–hnRNPU axis contributes to cell proliferation and drug resistance. Thus, targeting the CDC20-hnRNP-U-CTCF axis could be a novel strategy to neutralize the impaired chromatin condensation. It could inhibit tumor progression and resensitize TNBC to microtubule-targeting drugs.

## Figures and Tables

**Figure 1 cancers-14-03732-f001:**
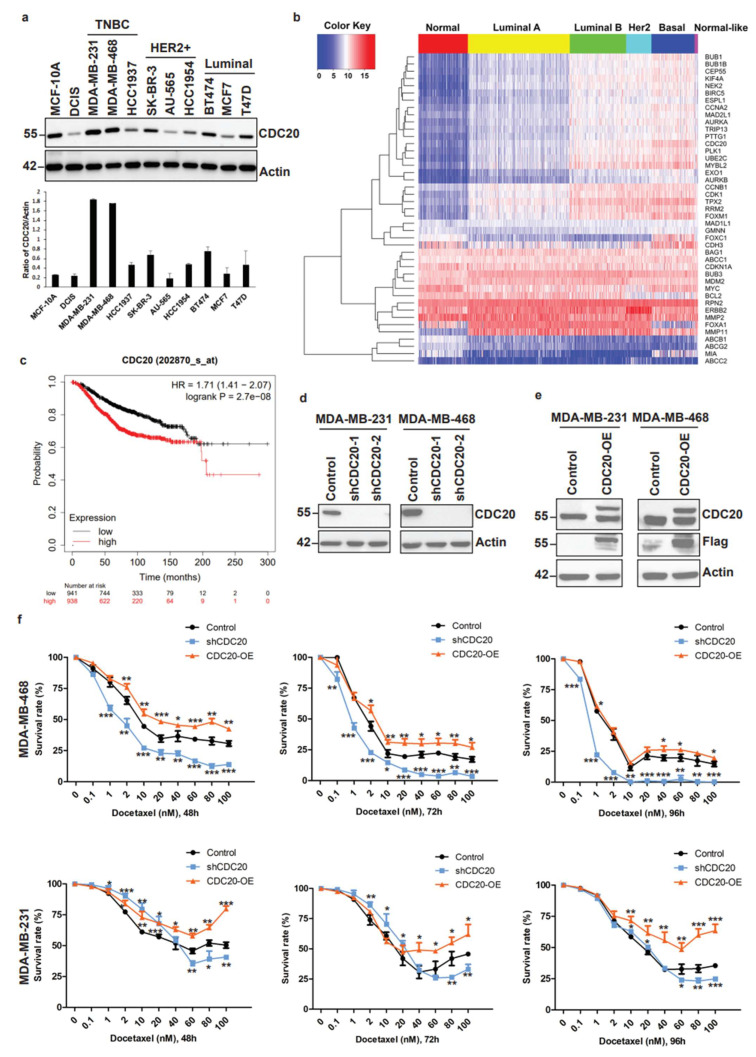
Elevated CDC20 correlates with poor breast cancer prognosis and drug resistance. (**a**) The expression of CDC20 was analyzed by Western blot in the normal breast epithelial cell line (MCF-10A), the ductal carcinoma in situ (DCIS) cell line, and different breast cancer cell lines. (**b**) Heatmap of the expression profile of CDC20, genome stability-, chromosome segregation-, mitotic exit-, drug resistance-, and CDC20-APC/C pathway-related genes. The transcriptome data from the Cancer Genome Atlas (TCGA) database were analyzed by hierarchical clustering. Red and blue colors represent higher and lower expression levels, respectively. (**c**) Kaplan–Meier overall survival (OS) curves of breast cancer patients with high and low CDC20 expression levels (Affymetrix ID: 202870_s_at). (**d**) MDA-MB-231 and MDA-MB-468 cells were transduced with either the scramble control shRNA or the CDC20 shRNAs #1 or #2 and processed by immunoblotting. (**e**) MDA-MB-231 and MDA-MB-468 cells were transduced with pHAGE-CDC20 plasmid to overexpress CDC20. The levels of CDC20 were measured by immunoblotting. (**f**) The survival percentage after CDC20 knockdown or overexpression in MDA-MB-468 and MDA-MB-231 cells was measured by CCK-8 cell survival assays after 48, 72, or 96 h of docetaxel treatment. Data are shown as mean ± SD for three independent experiments. *, *p* < 0.05; **, *p* < 0.01; ***, *p* < 0.001 versus Control.

**Figure 2 cancers-14-03732-f002:**
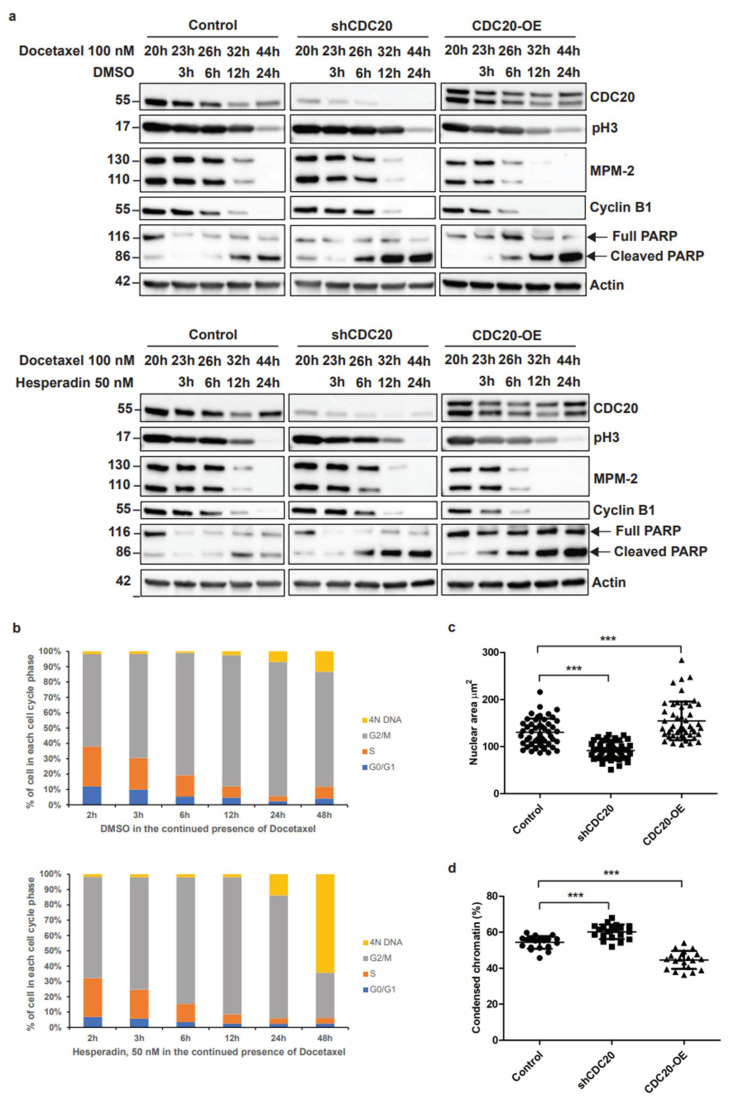
Elevated CDC20 is correlated with mitotic slippage, a possible reason causing an increase in nuclear size and chromatin decondensation in cancer cells. (**a**) MDA-MB-468 cells with CDC20 knockdown (shCDC20) or overexpression CDC20 (CDC20-OE) were pre-treated with 100 nM of docetaxel for 20 h. Pre-treated cells were cultured with either 50 nM of hesperadin or DMSO for 3, 6, 12, and 24 h in the presence of docetaxel. Immunoblot analysis was performed using antibodies against CDC20, pH3, MPM-2, Cyclin B1, PARP, and actin. The levels of pH3, cyclin B1, and MPM-2 were used as mitotic markers. (**b**) MDA-MB-468 cells were pre-treated with 100 nM of docetaxel for 20 h. Pre-treated cells were cultures with either 50 nM of hesperadin or DMSO for 3, 6, 12, 24, and 48 h in the presence of docetaxel. Cells were fixed with ice-cold ethanol, stained with propidium iodide staining solution, and analyzed by flow cytometry. Data are shown as the mean of two biological replicates. (**c**) Size of nuclei based on DAPI staining in MDA-MB-468 cells expressing CDC20 knockdown or overexpressing CDC20 (*n* = 50). (**d**) MDA-MB-468 cells expressing CDC20 knockdown or overexpressing CDC20 were transduced with mCherry-H2B. Cells were then analyzed by confocal microscopy to measure the percentage of condensed chromatin (*n* = 20). Data are shown as mean ± SD for three independent experiments. ***, *p* < 0.001.

**Figure 3 cancers-14-03732-f003:**
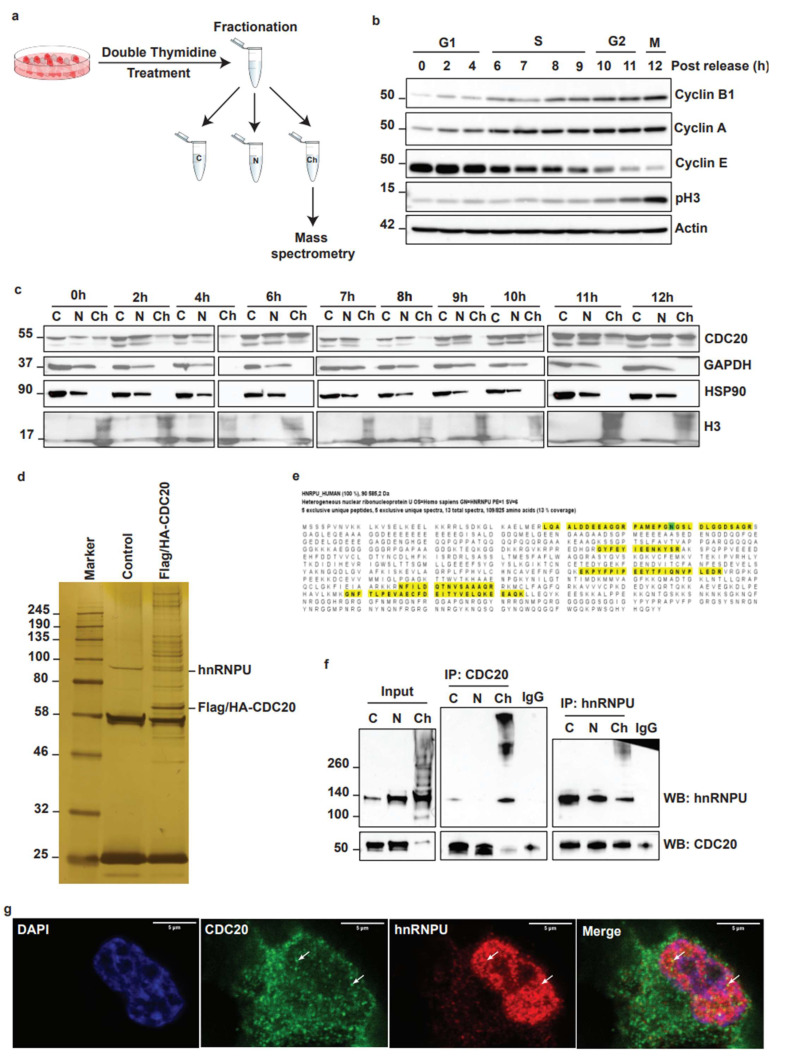
Identification of nuclear matrix protein hnRNPU as a target for CDC20 during chromatin dynamics. (**a**) Experimental Flow Chart (**b**) Validation of the efficiency of cell cycle synchronization in MDA-MB-468 cells. MDA-MB-468 cells were synchronized by double thymidine treatment and then released at different time points. The whole-cell lysates were subjected to Western blot analysis using the indicated antibodies. (**c**) Immunoblot analysis of CDC20 expression in different subcellular fractions through the cell cycle. MDA-MB-468 cells were synchronized by double thymidine treatment and then released at different time points, followed by DSP crosslinking. Crosslinked cells were fractionated into cytosolic (C), soluble nuclear (N), and chromatin (Ch) fractions. GAPDH and HSP90 were used as cytoplasmic markers, and histone 3 (H3) was used as a chromatin marker (**d**) Silver staining of Flag-M2 purification of the CDC20 protein complexes. Proteins that interact with CDC20 were purified from MDA-MB-468 cells expressing FLAG/HA-tagged CDC20 or MDA-MB-468 (control). Several proteins, including hnRNPU, were found in the CDC20 protein complex. (**e**) The sequences of mass spectrometry analysis for the identification of hnRNPU as an interacting partner of CDC20. The identified peptides were labeled in yellow. (**f**) Interaction between endogenous CDC20 and hnRNPU was validated by co-immunoprecipitation. (**g**) Validation of the interaction between CDC20 and hnRNPU by immunostaining and confocal microscopy in MDA-MB-231 breast cancer cells. Nuclei were stained using DAPI, CDC20 and hnRNPU were colocalized in the nucleus (white arrow).

**Figure 4 cancers-14-03732-f004:**
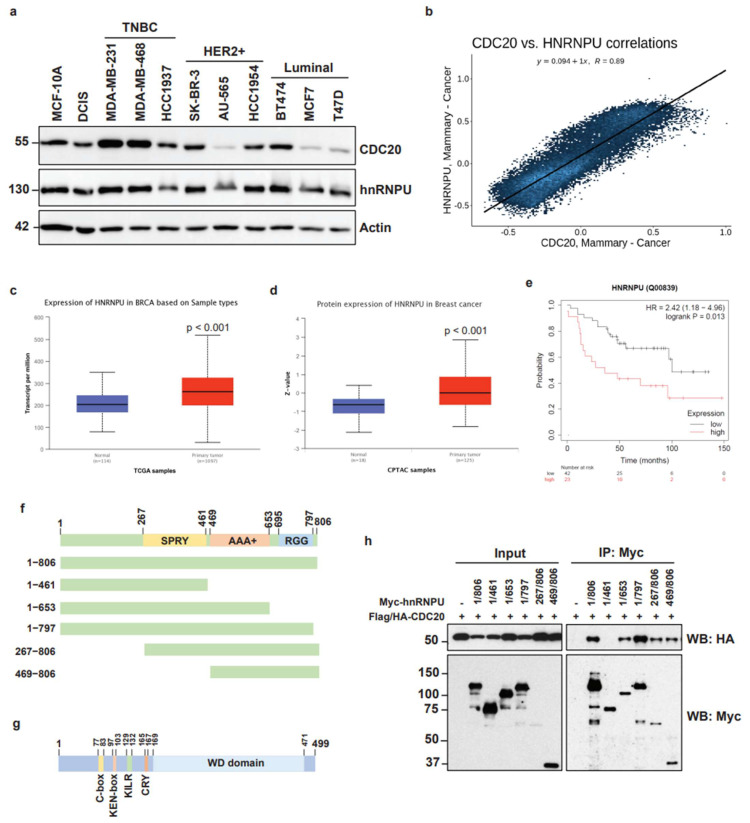
High expression of hnRNPU in breast cancer and mapping of molecular interaction between CDC20 and hnRNPU. (**a**) The expression of hnRNPU was analyzed by Western blot in the normal breast epithelial cell line (MCF-10A), the ductal carcinoma in situ (DCIS) cell line, and different breast cancer cell lines. (**b**) The correlation between CDC20 and hnRNPU expression in breast cancer patients was analyzed by using correlationAnalyzerR. (**c**) Box plots showing relative expression of hnRNPU mRNA in human breast cancer and normal tissues. (**d**) Box plots showing relative expression of hnRNPU protein in human breast cancer and normal tissues. The transcriptome and proteome data were obtained from TCGA and CPTAC databases, respectively, and box plots were generated using the UALCAN web resource. (**e**) Kaplan–Meier overall survival (OS) curves of breast cancer patients with high and low hnRNPU expression levels. (**f**) Schematic representation of human hnRNPU with previously identified domains and deletion constructs. (**g**) Schematic representation of human CDC20 with previously identified domains. (**h**) 293T cells were transfected with equal amounts of the indicated plasmids. After transfection for 48 h, cells were harvested, and the cell lysates were immunoprecipitated with an anti-Myc antibody. Immunoblot analysis revealed that the amino acid region 461–653 on hnRNPU is required for its interaction with CDC20.

**Figure 5 cancers-14-03732-f005:**
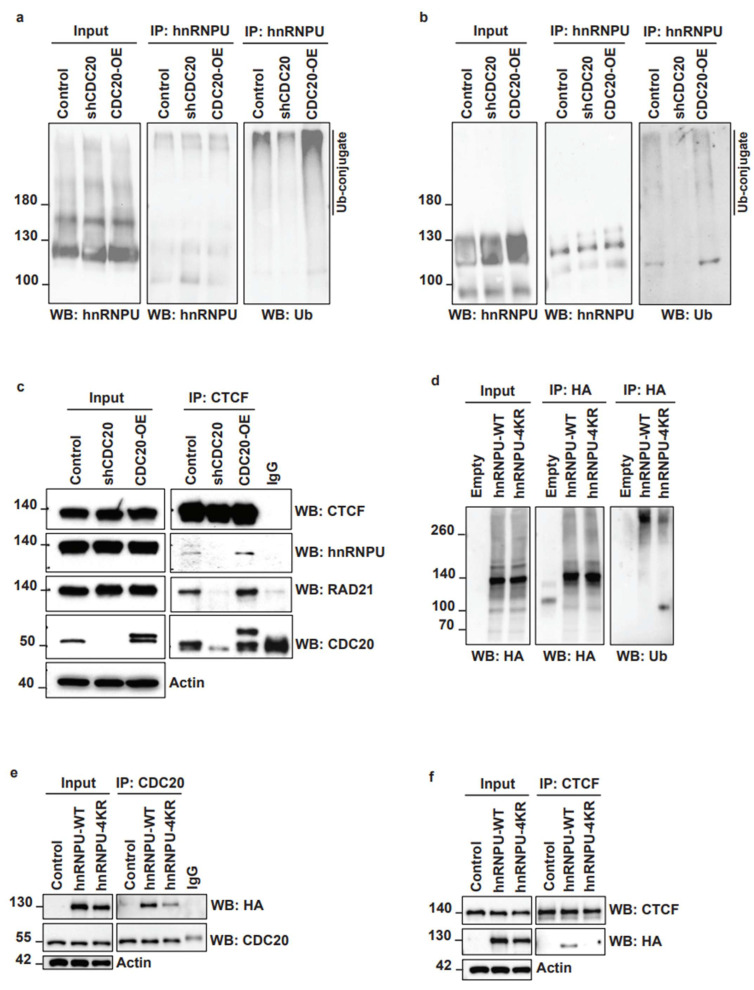
CDC20 regulates hnRNPU ubiquitination and its interaction with CTCF. (**a**,**b**) Ubiquitination of hnRNPU by CDC20. MDA-MB-468 control, expressing CDC20 knockdown or overexpressing CDC20 cells were crosslinked by using DSS (**a**) or not (**b**) before immunoprecipitation. The cell lysate was then immunoprecipitated with an anti-hnRNPU antibody. The level of ubiquitin was measured by immunoblot. (**c**) The cell lysate from MDA-MB-468 control, expressing CDC20 knockdown or overexpressing CDC20 cells was immunoprecipitated with an anti-CTCF antibody. The level of CTCF, hnRNPU, RAD21, and CDC20 was measured by immunoblot. (**d**) The cell lysate from MDA-MB-468 control, expressing hnRNPU-WT or hnRNPU-4KR was immunoprecipitated with an anti-V5 antibody. The level of ubiquitin was then measured by immunoblot. (**e**) The cell lysate from MDA-MB-468 control, expressing hnRNPU-WT or hnRNPU-4KR was immunoprecipitated with an anti-CDC20 antibody. The levels of HA-hnRNPU-WT or HA-hnRNPU-4KR were then measured by immunoblot. (**f**) The cell lysate from MDA-MB-468 control, expressing hnRNPU-WT or hnRNPU-4KR was immunoprecipitated with an anti-CTCF antibody. The levels of HA-hnRNPU-WT or HA-hnRNPU-4KR were then measured by immunoblot.

**Figure 6 cancers-14-03732-f006:**
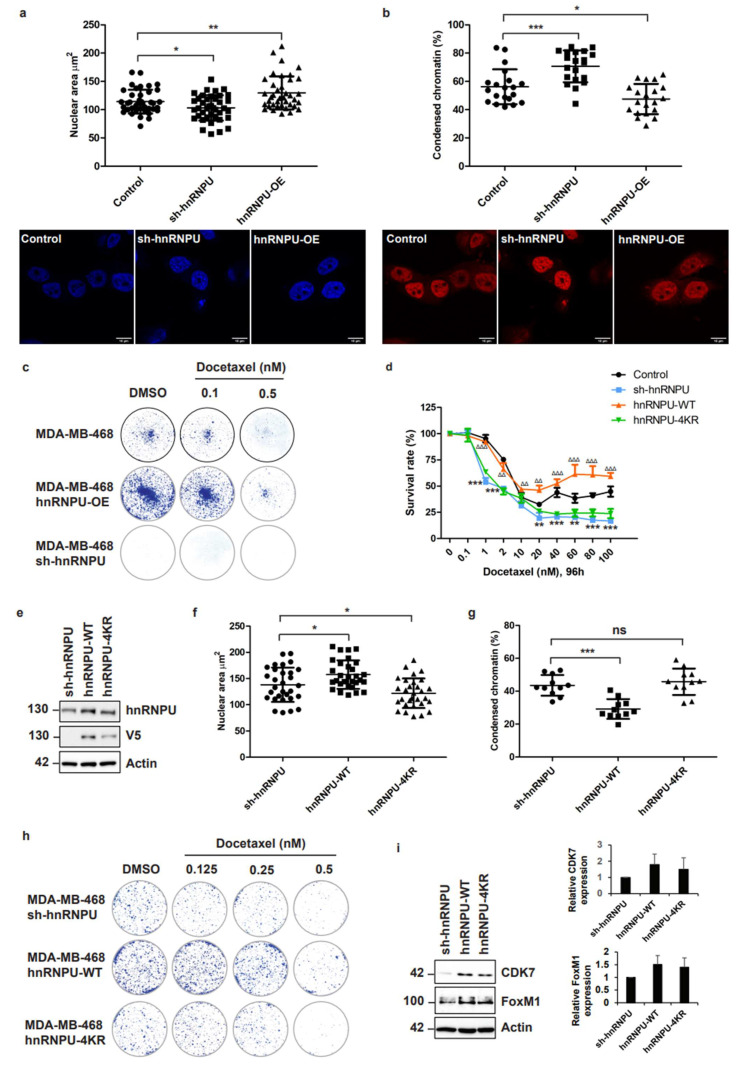
Inhibition of CDC20-mediated hnRNPU ubiquitination results in decreased nuclear area, enhanced chromatin condensation, and cellular sensitization to docetaxel treatment. (**a**) Size of nuclei based on DAPI staining in MDA-MB-468 cells expressing hnRNPU knockdown or overexpressing hnRNPU (*n* = 40). Representative images are shown. (**b**) MDA-MB-468 cells expressing hnRNPU knockdown or overexpressing hnRNPU were transduced with mCherry-H2B (red). Cells were then analyzed by confocal microscopy to measure the percentage of condensed chromatin (*n* = 20). Representative images are shown. (**c**) Clonogenic assay. MDA-MB-468 cells expressing hnRNPU knockdown or overexpressing hnRNPU were treated with 0.1 or 0.5 nM of docetaxel for 24 h. Then, the medium was changed and cells were cultured for 16 days in a drug-free medium. Colonies were fixed and stained with methanol/crystal violet. Colonies were photographed. (**d**) The survival percentage after hnRNPU knockdown, hnRNPU-WT rescue, or hnRNPU-4KR rescue in MDA-MB-468 cells was measured by CCK-8 cell survival assays after 96 h of docetaxel treatment. Data are shown as mean ± SD for three independent experiments. ∆∆, *p* < 0.01; ∆∆∆, *p* < 0.001 versus sh-hnRNPU. (**e**) MDA-MB-468 cells expressing hnRNPU knockdown were transduced with hnRNPU-WT or hnRNPU-4KR to rescue hnRNPU. The levels of hnRNPU-WT-V5 or hnRNPU-4KR-V5 were measured by immunoblotting. (**f**) Size of nuclei based on DAPI staining in MDA-MB-468 cells expressing hnRNPU knockdown, hnRNPU-WT, or hnRNPU-4KR (*n* = 30). (**g**) MDA-MB-468 cells expressing hnRNPU knockdown, hnRNPU-WT, or hnRNPU-4KR were transduced with mCherry-H2B. Cells were then analyzed by confocal microscopy to measure the percentage of condensed chromatin (*n* = 10). (**h**) Clonogenic assay. MDA-MB-468 cells expressing hnRNPU knockdown, hnRNPU-WT, or hnRNPU-4KR were treated with 0.125, 0.25, or 0.5 nM of docetaxel for 24 h. Then, the medium was changed and cells were cultured for 16 days in a drug-free medium. Colonies were fixed and stained with methanol/crystal violet. Colonies were photographed. (**i**) MDA-MB-468 cells expressing hnRNPU knockdown were transduced with hnRNPU-WT or hnRNPU-4KR to rescue hnRNPU. The levels of CDK7 and FoxM1 were measured by immunoblotting. All *p* values were defined as: *, *p* < 0.05; **, *p* < 0.01; ***, *p* < 0.001.

**Figure 7 cancers-14-03732-f007:**
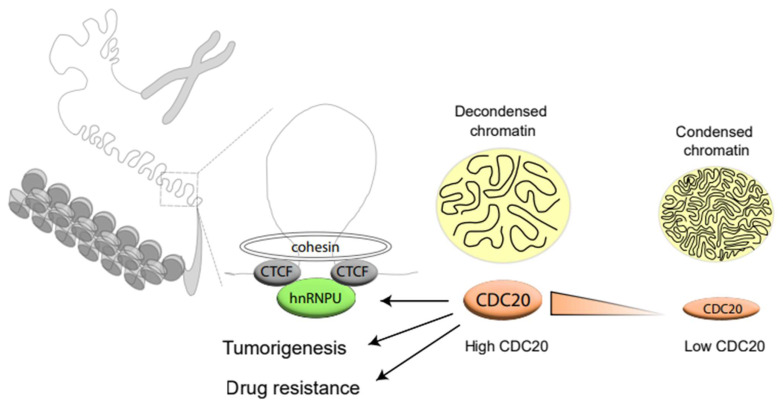
Aberrant CDC20–hnRNPU axis impairs the chromatin organization and causes drug resistance. Schematic diagram underlying the mechanism described in our study.

## Data Availability

All of the data generated or analyzed during this study are included in this article and the Appendix A. All data supporting the findings of this study are also available upon reasonable request from the corresponding author [Y.W.].

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
