# Peer review of "CDC20-Mediated hnRNPU Ubiquitination Regulates Chromatin Condensation and Anti-Cancer Drug Response"

_cancers, 2022, doi:10.3390/cancers14153732_

Round 1

Reviewer 1 Report

In the current manuscript, Wavelet-Vermuse et. al. demonstrated that CDC20-mediated hnRNPU ubiquitination modulates chromatin condensation by regulating the interaction between hnRNPU and CTCF-cohesin complex. The results look good and the data provided also supports the conclusion. The data is sound and novel as no such study has been reported that links CDC20 and hnRNPU. This study adds one more mechanism how of CDC20-hnRNPU axis regulates chromatin organization and drug resistance. The manuscript can be accepted in its current form.

Author Response

Comments and Suggestions for Authors

In the current manuscript, Wavelet-Vermuse et. al. demonstrated that CDC20-mediated hnRNPU ubiquitination modulates chromatin condensation by regulating the interaction between hnRNPU and CTCF-cohesin complex. The results look good and the data provided also supports the conclusion. The data is sound and novel as no such study has been reported that links CDC20 and hnRNPU. This study adds one more mechanism how of CDC20-hnRNPU axis regulates chromatin organization and drug resistance. The manuscript can be accepted in its current form.

Thank you for reviewing our manuscript. We greatly appreciate your feedback. We have proofread the manuscript for language and grammatical errors. We have tracked all modifications.

Reviewer 2 Report

In this paper, authors find out the upregulation of CDC20 in cancer cells and the upregulation of CDC20 linked to drug resistance. Numerous processes, including alterations in the dynamic and organizational structure of chromatin, can lead to drug resistance. So they checked that the hnRNPU is a binding partner of CDC20 and it is responsible to upregulate the CDC20. In this manuscript they did numerous experiments like co immunoprecipitation, immunostaining, clonogenic assay to check the CDC20-mediated hnRNPU ubiquitination, effects of CDC20-hnRNPU on nuclear size and chromatin condensation, role of CDC20-hnRNPU in tumor progression and drug resistance. In short, they concluded that the dysregulation of the CDC20-hnRNPU axis contributes to tumor progression and drug resistance.

Questions

1.    In figure 2, the authors used hesperidin to treat cells, however; the rational is missed. Authors should address more information in this part.

2.    The information of hnRNPU/CTCF regulation should be addressed in the manuscript.

3.    In this manuscript author need to address the correlation between mitotic slippage mechanism and drug resistance.

4.    Authors mentioned that CDC20 has a function on drug resistance. Whether CDC20/hnRNPU pathway has a critical role in docetaxel resistant cells?

5.    In figure 5, authors demonstrated that CDC20 regulates hnRNPU ubiquitination. It will be interesting to show whether proteasome inhibitors or autophagy inhibitors interrupt CDC20-mediated hnRNPU degradation?

6.    Authors showed that overexpression of CDC20 induces hnRNPU ubiquitination. However, authors should further investigate the degradation level of hnRNPU. Moreover, whether knock down of CDC20 can reverse the hnRNPU expression upon docetaxel treatment?

Author Response

Comments and Suggestions for Authors

In this paper, authors find out the upregulation of CDC20 in cancer cells and the upregulation of CDC20 linked to drug resistance. Numerous processes, including alterations in the dynamic and organizational structure of chromatin, can lead to drug resistance. So they checked that the hnRNPU is a binding partner of CDC20 and it is responsible to upregulate the CDC20. In this manuscript they did numerous experiments like co immunoprecipitation, immunostaining, clonogenic assay to check the CDC20-mediated hnRNPU ubiquitination, effects of CDC20-hnRNPU on nuclear size and chromatin condensation, role of CDC20-hnRNPU in tumor progression and drug resistance. In short, they concluded that the dysregulation of the CDC20-hnRNPU axis contributes to tumor progression and drug resistance.

Thank you for reviewing our manuscript and for all your comments and suggestions. We have proofread the manuscript for language, grammar errors, and clarity. We have tracked all corrections to show any changes and all edits. We have added some additional information to the Introduction part. We also have added some information to clarify the rationale of some experiments. 

Questions

  1. In figure 2, the authors used hesperidin to treat cells, however; the rational is missed. Authors should address more information in this part.

We explained the rationale for the hesperadin treatment used in Figure 2a (Results, Page 9 Lines 297-302). In addition, we modified the labeling of the figure 2a top panel by adding DMSO treatment and time points for clarification of the experimental design.

  1. The information of hnRNPU/CTCF regulation should be addressed in the manuscript.

The information about hnRNPU/CTCF regulation is limited since only a few papers were published. We added more information about the hnRNPU/CTCF regulation reported previously in the Introduction (Page 4 Lines 102-107) and the Discussion (Page 22 Lines 569-573; 586-596) sections.

  1. In this manuscript author need to address the correlation between mitotic slippage mechanism and drug resistance.

The mitotic slippage is one of the possible mechanisms of drug resistance, where cancer cells can escape from taxane (eg. Docetaxel, Paclitaxel)- induced mitotic arrest and exit mitosis to the next interphase. We added more information about the correlation between mitotic slippage and drug resistance. In particular, we described the role of Aurora B kinase inhibitors and spindle assembly checkpoint (SAC) in mitotic slippage and how it is associated with drug resistance (Page 9 Lines 286-300).

  1. Authors mentioned that CDC20 has a function on drug resistance. Whether CDC20/hnRNPU pathway has a critical role in docetaxel resistant cells?

CDC20 function in drug resistance has been shown previously. The CDC20/hnRNPU pathway is new and has not been described yet. In this paper, we showed the role of CDC20 in drug resistance developed by TNBC. As shown in Figure 1f, the knockdown of CDC20 sensitizes the TNBC cells to docetaxel treatment, however, the overexpression of CDC20 increases the resistance of TNBC cells to docetaxel treatment (Results part: Page 9 Lines 272-274). Additionally, In Figures 6c and d, we showed that the knockdown of hnRNPU sensitizes the TNBC cells to docetaxel treatment, however, the overexpression of hnRNPU increased the resistance of the cells to docetaxel treatment (Results part: Page 18 Lines 499-502). Moreover, in Figure 6d and h, we showed that hnRNPU-4KR mutant had similar effects as hnRNPU knockdown in sensitizing TNBC cells to docetaxel treatment which suggests that CDC20-mediated hnRNPU ubiquitination plays an important role in docetaxel resistance (Results part: Page 18 Line 512-514).   

  1. In figure 5, authors demonstrated that CDC20 regulates hnRNPU ubiquitination. It will be interesting to show whether proteasome inhibitors or autophagy inhibitors interrupt CDC20-mediated hnRNPU degradation?

In Supplementary Figure S4a, we analyzed whether CDC20 regulates the stability of hnRNPU protein by treating the cells with cycloheximide, a protein synthesis inhibitor. To our surprise, neither CDC20 knockdown nor CDC20 overexpression affects the protein stability of hnRNPU. Moreover, in Supplementary Figure S4b, we treated the MDA-MB-468 control cells, cells with CDC20 knockdown or overexpression with MG132 (proteasome inhibitor), chloroquine (autophagy inhibitor), or DMSO and analyzed the hnRNPU expression levels by immunoblot. Any significant differences were observed between DMSO and MG132 treatment and between DMSO and chloroquine treatment. We added several sentences in the Results (Page 16 Lines 442-444) and Discussion parts (Page 22 Lines 588-596). Thus, CDC20-mediated hnRNPU ubiquitination is not for degradation, but probably, to help/drive the interaction between hnRNPU and chromatin and the formation of the CTCF-RAD21 complex.

  1. Authors showed that overexpression of CDC20 induces hnRNPU ubiquitination. However, authors should further investigate the degradation level of hnRNPU. Moreover, whether knock down of CDC20 can reverse the hnRNPU expression upon docetaxel treatment?

As we explained in question 5, our data suggest that CDC20-mediated hnRNPU ubiquitination does not target hnRNPU for degradation but rather may play an important regulatory role in the interaction between hnRNPU and chromatin and between hnRNPU and other proteins, such as CTCF or RAD21, to regulate chromatin loop formation and 3D chromatin organization. We investigated the degradation level of hnRNPU by using proteasomal or autophagy inhibitors (supplemental figure S4b), however, we didn’t see any significant difference in hnRNPU levels. For these reasons, we didn’t measure the hnRNPU levels in breast cancer cells expressing CDC20 knockdown upon docetaxel treatment.

Reviewer 3 Report

Major Concerns

1.) Images are of very low resolution.  It is very difficult to assess the quality of the data with the images at such low resolution.  Please upload higher resolution images so that I can review them properly for content.

2.) Data from supplemental figure 2 must be done several times to provide biological replicates. The aggregated data, including stats on the newly presented replicates, should be included in Figure 2.  The measurement of 4N DNA content is an important piece of data to suggest the accumulation of cells undergoing slippage. The % in each cell cycle phase and ploydy can be summarized in a table with a few representative traces.

3.) None of the pulldowns were done with a DNase or RNase treatments. When pulling down proteins from the nuclear fraction it is important to control for DNA and RNA bridging.  It is very possible that the physical interactions observed are through nucleic acid bridging. Please repeat the key pulldowns, but with an excessive DNase or RNase treatment to determine if the pull down required nucleic acids.

4.) Please repeat the studies in Figure 6 in the MDA-MB-231 cell line.  Ans also a "normal" MCF10A or IMR90 cell line model.  If the authors are claiming that these observations will provide a vulnerability to exploit in cancer cells, then they should not be observed in normal cells, otherwise the treatment will be equally toxic to the patients' normal cells.

Minor Concerns

1.) Please remove any reference to the observations relating to tumorgenicity.  No in vivo tumor studies are done, so these studies can only be extrapolated to cancer cell growth.  No tumors are studied and therefore the observations cannot be extrapolated to tumor growth.

2.) Please remove the claim that the observations are affecting chromosome looping - lines 408 to 410. No measurements of looping have been made via 3C or similar technique. CTCF has a variety of functions, one of which is promoting looping, and this is speculative and should be saved for the discussion.  Also, in the discussion please reference that the interactions could affect CTCF in its variety of functions in gene regulation.

Author Response

Comments and Suggestions for Authors

Major Concerns

1.) Images are of very low resolution.  It is very difficult to assess the quality of the data with the images at such low resolution.  Please upload higher resolution images so that I can review them properly for content.

We uploaded higher-resolution images.

2.) Data from supplemental figure 2 must be done several times to provide biological replicates. The aggregated data, including stats on the newly presented replicates, should be included in Figure 2.  The measurement of 4N DNA content is an important piece of data to suggest the accumulation of cells undergoing slippage. The % in each cell cycle phase and ploidy can be summarized in a table with a few representative traces.

Supplemental figure 2 shows representative flow cytometry histograms of two biological replicates. The aggregated data were included in Figure 2b. We also updated figure legends for supplemental figure 2 and figure 2b. Statistical analysis were added in supplemental figure S3a. Thank you for your great suggestion.

3.) None of the pulldowns were done with a DNase or RNase treatments. When pulling down proteins from the nuclear fraction it is important to control for DNA and RNA bridging.  It is very possible that the physical interactions observed are through nucleic acid bridging. Please repeat the key pulldowns, but with an excessive DNase or RNase treatment to determine if the pull down required nucleic acids.

I agree that it is important to control for DNA and RNA bridging. When we purified the Flag/HA-CDC20 protein complexes (Figure 3d), we used micrococcal nuclease which digests single-stranded and double-stranded DNA and RNA. The physical interactions observed don’t seem to be through the nucleic acid bridging. For this reason, we didn’t use DNase or RNase treatments in future pulldowns.

4.) Please repeat the studies in Figure 6 in the MDA-MB-231 cell line.  Ans also a "normal" MCF10A or IMR90 cell line model.  If the authors are claiming that these observations will provide a vulnerability to exploit in cancer cells, then they should not be observed in normal cells, otherwise the treatment will be equally toxic to the patients' normal cells.

I agree that repeating the studies in Figure 6 in the MDA-MB-231 cell line would confirm the observations made in the MDA-MD-468 cell line, however, it is not doable due to the time allowed for revisions. The development of resistance to chemotherapy like docetaxel is a major clinical problem in human cancers including triple negative breast cancer patients. The purpose of our study was to understand why some breast cancer cells, particularly TNBC, are resistant to docetaxel. For this reason, we only used triple negative breast cancer cells in our study.

Minor Concerns

1.) Please remove any reference to the observations relating to tumorgenicity.  No in vivo tumor studies are done, so these studies can only be extrapolated to cancer cell growth.  No tumors are studied and therefore the observations cannot be extrapolated to tumor growth.

References to the observations relating to tumorigenicity were removed (Discussion, Page 21-22 Line 560 and 633, Conclusions Page 24, Line 643)

2.) Please remove the claim that the observations are affecting chromosome looping - lines 408 to 410. No measurements of looping have been made via 3C or similar technique. CTCF has a variety of functions, one of which is promoting looping, and this is speculative and should be saved for the discussion.  Also, in the discussion please reference that the interactions could affect CTCF in its variety of functions in gene regulation.

We edited lines 408 to 410. Also, in the discussion, we referenced that the interactions can affect CTCF in its variety of functions in gene regulation (Page 21 Lines 569-573)

Reviewer 4 Report

In this manuscript Wavelet-Vermuse et. al., highlight the role of CDC20-mediated hnRNPU ubiquitination in modulating chromatin condensation through the CTCF-cohesin complex. Dysregulation of the CDC20-hnRNPU axis contributes to tumor progression and drug resistance. This is an interesting paper in which the authors examined the effects of CDC20 knockdown and Overexpression, as well as its binding partner, hnRNPU, on chromatin dynamics and response to docetaxel treatment. 

In general, the quality of the images needs to be improved (especially the confocal images).

More specific comments:

Figure 1b.: the authors should improve the quality of the Figure. We can not see the name of the genes in the heat map.

Figure 2a: Quantification of the western blot bands. There is no difference between the control and the shCDC20 group (regarding the pH3, MPM-2 and cyclin B1) but only between the control/sh and the CDC20-OE group. The authors should make a comment about that. Please show which is the full length vs the cleaved PARP band and explain in more detail the differences between the groups/conditions. 

Figure 2b: Low resolution of the confocal images. Please improve them.

Figure 3e: Improve the quality of the image.

Figure 4: Have the authors checked the correlation between the CDC20 and hnRNPU expression in breast cancer patients? Comparison of this correlation between the luminal and TNBC subtypes.  

Figure 5a,b: Please explain the difference between the two experiments a vs b.

Figure 6a,b: Improve the quality of the confocal images.

Figure 6c,h: 1) Based on the these figures there is a difference in the number of colonies between the control and the hnRNPU-OE or KD cells in the non treated groups (DMSO). The same in Fig. 6h between the hnRNPU KD and the WT/4KR rescue groups. (The control group is missing). The authors should provide an explanation about this difference in the non treated conditions. Is this difference due to the lower rate of proliferation or due to increased cell death in the RNPU KD or 4KR mutant cells? Apoptotic cell death is characterised by chromatin condensation. Could the authors provide Annexin V staining data before and after the docetaxel treatment?

2) Chromatin condensation is also associated with the response against other chemotherapeutic drugs. Have the authors tried to repeat their experiments using e.g. doxorubicin? 

Author Response

Comments and Suggestions for Authors

In this manuscript Wavelet-Vermuse et. al., highlight the role of CDC20-mediated hnRNPU ubiquitination in modulating chromatin condensation through the CTCF-cohesin complex. Dysregulation of the CDC20-hnRNPU axis contributes to tumor progression and drug resistance. This is an interesting paper in which the authors examined the effects of CDC20 knockdown and Overexpression, as well as its binding partner, hnRNPU, on chromatin dynamics and response to docetaxel treatment. 

In general, the quality of the images needs to be improved (especially the confocal images).

Thank you for your suggestions and for reviewing our work! We have proofread the manuscript for any English language errors and typos. We have edited some parts of the manuscript and highlighted all edits to further allow an easier read. We also uploaded all figures with high resolution and quality during revision submission.

More specific comments:

Figure 1b.: the authors should improve the quality of the Figure. We cannot see the name of the genes in the heat map.

The quality of the Figure was improved.

Figure 2a: Quantification of the western blot bands. There is no difference between the control and the shCDC20 group (regarding the pH3, MPM-2 and cyclin B1) but only between the control/sh and the CDC20-OE group. The authors should make a comment about that. Please show which is the full length vs the cleaved PARP band and explain in more detail the differences between the groups/conditions. 

We added several sentences to explain why we didn’t see significant differences in the pH3, MPM-2, and cyclin B1 levels in the Results Part (Page 9 Lines 288 to 297). In figure 2a, we labeled which band is the full-length vs the cleaved PARP. We also explained in more detail the differences between the groups in the Results part (Page 10, Line 313 to 317).

Figure 2b: Low resolution of the confocal images. Please improve them.

The quality of the confocal images was improved

Figure 3e: Improve the quality of the image.

The quality of the image was improved

Figure 4: Have the authors checked the correlation between the CDC20 and hnRNPU expression in breast cancer patients? Comparison of this correlation between the luminal and TNBC subtypes.

In Figure 4a, we show the expression of CDC20 and hnRNPU in human breast cancer cell panel, where we can see positive correlation. In addition, we can show the correlation between the CDC20 and hnRNPU expression in breast cancer patients done by using correlationAnalyzerR (we added the results in Figure 4b). The analysis was done in all breast cancer subtypes since in figure 4a we show a correlation in different cell lines which is subtype-independent. 

Figure 5a,b: Please explain the difference between the two experiments a vs b.

The cell lysate immunoprecipitated with anti-hnRNPU antibody was prepared from cells crosslinked by using DSS (disuccinimidyl suberate) or not, Figure 5a and 5b, respectively. Crosslinking with DSS was used to preserve hnRNPU oligomers and determine whether ubiquitination happens on hnRNPU monomers and/or oligomers. To make sure that the results observed in figure 5a do not result from non-specific bound proteins we repeated the experiment without a crosslinker in figure 5b. For clarity, we updated the figure legends for figures 5a, b. 

Figure 6a,b: Improve the quality of the confocal images.

The quality of the confocal images was improved

Figure 6c,h: 1) Based on the these figures there is a difference in the number of colonies between the control and the hnRNPU-OE or KD cells in the non treated groups (DMSO). The same in Fig. 6h between the hnRNPU KD and the WT/4KR rescue groups. (The control group is missing). The authors should provide an explanation about this difference in the non treated conditions. Is this difference due to the lower rate of proliferation or due to increased cell death in the RNPU KD or 4KR mutant cells? Apoptotic cell death is characterised by chromatin condensation. Could the authors provide Annexin V staining data before and after the docetaxel treatment?

We measured the cell proliferation of MDA-MB-468 control, hnRNPU knockdown, and overexpression by using CCK-8. We found that the cells overexpressing hnRNPU grow faster than control cells. In contrast, the cells expressing hnRNPU knockdown grow slower than control cells. We updated the Results part in the main text (Page 18 Lines 502 to 505) to include and interpret these results. We also included these results in Supplementary Figure S5c. In Figure 6h, we performed the clonogenic assay to study the effect of hnRNPU-WT and hnRNPU-4KR on colony forming ability under docetaxel treatment in MDA-MB-468 cells expressing hnRNPU knockdown. Thus, we used MDA-MB-468 expressing hnRNPU knockdown as control which has the same genetic background as experimental cell lines (hnRNPU-WT and -4KR were ectopically expressed in sh-hnRNPU cells).

2) Chromatin condensation is also associated with the response against other chemotherapeutic drugs. Have the authors tried to repeat their experiments using e.g. doxorubicin?

At the beginning of our study, we tried several drugs, like Docetaxel, Rapamycin, Olaparib, and Palbociclib, which showed more or less effect on the killing of TNBC cells. We decided to continue our study with Docetaxel since numerous studies showed that the level of CDC20 influences the response of cancer cells to Docetaxel treatment. Moreover, the mechanism by which cancer cells overexpressing CDC20 develop resistance to Docetaxel treatment was largely unknown.

Round 2

Reviewer 2 Report

The authors have addressed my comments and I am satisfied with the improvements.

Reviewer 3 Report

The reviewers have addressed the comments.